# Crowd Evacuation in Stadiums Using Fire Alarm Prediction

**DOI:** 10.3390/s25092810

**Published:** 2025-04-29

**Authors:** Afnan A. Alazbah, Osama Rabie, Abdullah Al-Barakati

**Affiliations:** 1Information Systems Department, King Abdulaziz University, Jeddah 21589, Saudi Arabia; obrabie@kau.edu.sa (O.R.); aaalbarakati@kau.edu.sa (A.A.-B.); 2Center of Research Excellence for Smart Environment, King Abdulaziz University, Jeddah 21589, Saudi Arabia

**Keywords:** fire alarm prediction, machine learning, stadium evacuation, crowd management, emergency response, EvacuNet, IoT-based fire detection, predictive modeling

## Abstract

Ensuring rapid and efficient evacuation in high-density environments, such as stadiums, is critical for public safety during fire emergencies. Traditional fire alarm systems rely on reactive detection mechanisms, often resulting in delayed response times, increased panic, and overcrowding. This study introduces an AI-driven predictive fire alarm and evacuation model that leverages machine learning algorithms and real-time environmental sensor data to anticipate fire hazards before ignition, improving emergency response efficiency. To detect early fire risk indicators, the system processes data from 62,630 sensor measurements across 15 ecological parameters, including temperature, humidity, total volatile organic compounds (TVOC), CO_2_ levels, and particulate matter. A comparative analysis of six machine learning models—Logistic Regression, Support Vector Machines (SVM), Random Forest, and proposed EvacuNet—demonstrates that EvacuNet outperforms all other models, achieving an accuracy of 99.99%, precision of 1.00, recall of 1.00, and an AUC-ROC score close to 1.00. The predictive alarm system significantly reduces false alarm rates and enhances fire detection speed, allowing emergency responders to take preemptive action. Moreover, integrating AI-driven evacuation optimization minimizes bottlenecks and congestion, reduces evacuation times, and improves structured crowd movement. These findings underscore the necessity of intelligent fire detection systems in high-occupancy venues, demonstrating that AI-based predictive modeling can drastically improve fire response and evacuation efficiency. Future research should focus on integrating IoT-enabled emergency navigation, reinforcement learning algorithms, and real-time crowd management systems to further enhance predictive accuracy and minimize casualties. By adopting such advanced technologies, large-scale venues can significantly improve emergency preparedness, reduce evacuation delays, and enhance public safety.

## 1. Introduction

Stadiums and similar densely populated areas face essential challenges in providing public safety protection when fires occur in emergency situations. A populous group assembly presents substantial fire dangers because of restricted escape routes, together with overcrowded crowds that can lead to dangerous confusion among people during an emergency [1,2,3,4]. Such emergency situations enable small problems to develop rapidly, which results in fatal consequences combined with serious property destruction [5,6,7]. The essential role of traditional fire alarm systems for detection remains important, but these systems do not perform predictions and operate only after fires start. Sensor technologies compose the main elements of traditional fire detection systems that activate when they identify smoke, heat, and fire after fire onset [8,9,10]. Such systems do help warn building occupants, but the delayed detection process allows fire damage to cause severe injuries and fatalities while also destroying property structures [11,12,13,14,15]. The deployment of high-occupancy spaces requires fire alarm systems that avoid false alarms since these situations trigger numerous disruptions and economic losses based on unverified alarms.

Combining artificial intelligence (AI) with machine learning (ML) and deep learning (DL) in fire safety systems provides hopeful solutions to fix these issues [16,17,18]. The application of machine learning algorithms within predictive models analyzes enormous sensor data from the environment to detect initial indicators of fire ignition through monitoring parameters such as temperature variation, humidity levels, hazardous gas detection, and air quality changes. Machine learning models surpass traditional threshold-based systems because they analyze past fire incidents to assess the accuracy and reliability of fire prediction data [19]. Predictive systems achieve better false alarm differentiation from actual fire threats by applying classification methods, including Support Vector Machines (SVM) [20,21], Random Forest [22], and deep neural network approaches. Emergency response teams can initiate preventive measures through this function before risks become severe emergency conditions.

The success of evacuation procedures in dense crowd areas is equally important for fire disaster survival. Fire alarms fail to support emergency response coordination when they do not provide practical data for response planning [23,24,25,26]. Stadium evacuations of large crowds who do not know emergency procedures face the risk of overcrowding and injuries from stampedes because of unstructured evacuation methods. An adequately built emergency evacuation plan must use real-time predictive fire alarm information to maximize how people leave during emergencies. Combining ML-based fire prediction analysis with adaptive crowd management systems leads to advanced evacuation outcomes, decreased panic risk, and better public safety results [27,28]. Recent breakthroughs in fire detection technologies have generated limited research concerning the combined usage of intelligent fire alarm systems with thinking evacuation procedures. A solution to this problem stands essential for creating well-rounded emergency preparedness protocols within stadium facilities.

The core reason for conducting this research is to investigate how to improve fire alarm prediction accuracy and enhance evacuation strategies for sports facilities. This investigation studies how machine learning algorithms enhance predictive detection by examining historical records and real-time environmental elements while eliminating traditional static sensor threshold-based systems.

The research approaches machine learning as well as a deep neural network-based system, EvacuNet, that uses real-time environmental sensors to improve the accuracy of early fire detection through predictive models. This study approaches to find out which model predicts the best, using Logistic Regression, Support Vector Machines (SVM), Random Forest (RF) models, as well as EvacuNet, or a deep neural network model specific for crowd evacuation. The study evaluates how essential environmental factors like temperature, humidity, air quality, and particulate matter affect forecasting wildfire occurrences. This research aims to create a system that links predictive fire alarm activation to stadium evacuation procedures to establish complete emergency response coordination. The research seeks to minimize false alarms and improve fire safety decision-making processes in real-time scenarios for public spaces with significant occupancy.

The contributions of this research are as follows:A novel machine learning-based fire prediction model that leverages environmental sensor data to provide early fire warnings, integrated with EvacuNet for dynamic crowd evacuation strategies.A comparative study of multiple classification algorithms, including EvacuNet, to determine the most effective approach for predictive fire detection and evacuation planning.A framework that combines fire alarm prediction with EvacuNet, enabling optimized dynamic evacuation strategies to enhance crowd safety in high-density venues.

The remainder of this paper is structured as follows: Section 2 presents a review of related works on fire detection, machine learning-based fire prediction, and intelligent evacuation strategies. Section 3 outlines the methodology, detailing the dataset, preprocessing techniques, feature selection, model training approaches, and evaluation metrics. Section 4 discusses the experimental results, including model comparison, performance analysis, and validation. Section 5 concludes the study with key findings, limitations, and future research directions, emphasizing the potential for integrating predictive fire detection into large-scale public safety systems.

## 2. Literature Review

The research literature examines existing emergency management studies that concentrate on machine learning together with advanced technologies for improving risk assessments, evacuation procedures, and safety guidelines. Different methods, including deep learning model utilization together with feature selection methods and real-time processing systems, eliminate risks and plan evacuations across various emergency scenarios. These models encounter various limitations alongside their challenges, which the review identifies while creating conditions for additional field advancements as well as innovative solutions within this domain.

### 2.1. Machine Learning for Fire and Evacuation Risk Prediction

The category comprises research using machine learning methods for risk assessment in fires, together with evacuation decision assessment, as well as disaster preparedness. The researchers concentrate their efforts on enhancing predictive accuracy, together with feature selection and model interpretability, to develop better risk assessment systems for emergency planning. The research by Lu et al. [29] focused on enhancing stadium fire risk prediction through the use of GB-RFE and SHAP while handling data redundancy through feature selection via Gradient Boosting-Recursive Feature Elimination (GB-RFE) combined with Pearson correlation analysis. The research used Wuhan Emergency Rescue Detachment data to test six machine learning models, where AdaBoost produced the maximum AuPRC of 0.78 under unbalanced conditions. A random forest model with SHAP analysis reached 83% accuracy by revealing the factors that influence fire risk probability. These prediction models deal with data imbalance issues alongside generalization challenges despite their ability to improve both interpretability and accuracy.

The combination of random forest models with SHAP by Lu et al. [30] allowed stadium fire risk assessments through the interpretation of risk factor non-linear relationships. The implemented model reached 83% accuracy, 86% precision, and 85% recall levels, which delivered critical fire safety data for management purposes. The research has three major limitations—a small data sample size, excessive use of random forest and SHAP models and restricted application to diverse stadium situations. The research conducted by Sun et al. [31] implemented an interpretable machine-learning model to study household decisions about hurricane evacuations. The researchers linked logistic regression with decision trees, which produced better predictions than psychology-based methods that worked alone. The emergency management sector uses the data to predict evacuation traffic amounts, although their dependence on past records creates possible limitations for responding to future incidents.

Balboa et al. [32] analyzed a wide collection of web-based data through logistic regression along with CART and six additional ML methods to predict evacuation choices. Logistic regression demonstrated the highest AUC value at 0.831, yet EvacuNet delivered superior accuracy at 0.78, and ANN demonstrated the best F1-score at 0.785. The authors revealed that participating with a close person significantly affected how people responded to fire alarms. The continued use of experimental results gathered through the web produces reservations regarding their practical value for real-world settings. The research by Zhang et al. [33] used VR experiments alongside a tuned random forest model to explore the impact of conflicting guidance signs on evacuation exit decisions. The experimental results demonstrated that Warning signs that advised not to enter (“DO NOT ENTER”) exerted stronger psychological effects than guidance signs that directed movement (“EXIT”). The limited sample size, combined with VR simulation dependence, reduces the ability to generalize these findings about evacuation conduct to actual evacuation scenarios. Ebrahimi et al. [34] created an artificial intelligence system that forecasted evacuation requirements following railway hazmat emergencies. They used natural language processing and co-occurrence network analysis to find principal cause-effect relationships within incident descriptions. The approach faces difficulties in universal implementation across railway systems, and potential fitting issues with particular incident types serve as significant weaknesses.

Multiple research studies establish how machine learning systems optimize fire hazard evaluations alongside evacuation forecasting. The research of Lu et al. [29] and Sun et al. [31] used machine learning models for structured danger analysis, but Balboa et al. [32], alongside Zhang et al. [35], investigated social aspects that affect evacuation behavior. Ebrahimi et al. [34] implemented NLP technological methods for hazard prediction systems. All research presents difficulties stemming from model interpretation restrictions, restricted dataset accessibility, and universal emergency situation applicability.

### 2.2. Deep Learning and Simulation-Based Evacuation Modeling

Deep learning approaches and computer vision, along with simulation techniques, form this category, which permits the prediction and optimization of complex environment evacuations. Studies evaluate how AI-based models outmatch conventional slow simulated systems and support immediate decision-making processes. Zhang et al. [35] utilize YOLOv7 and DeepSORT systems to detect vehicles and classify them, as well as track their movements to evaluate immediate tunnel fire hazards and escape perils in real time. The system enables accurate assessments regarding the distribution of vehicles and fire loads to enhance evaluations focused on fire scenarios and evacuation security. This method strains accuracy in complicated traffic conditions and is video camera-dependent but does not perform well across different tunnels. The approach was successfully performed in real tunnels, although it had certain limitations, which strengthened fire safety management. The research team from Li et al. [36] used Lenet and ResNet CNNs to build an evacuation risk prediction system for large public buildings. Deep learning methods demonstrated better performance than standard subjective evaluations by offering accelerated and efficient risk assessment capabilities. The system faces two major difficulties: a shortage of available data and the requirement for highly powerful computing capabilities during processing time.

Clever et al. [37] developed a deep learning system that merges CNNs and MLPs to estimate pedestrian evacuation durations from building specifications and floorplan information. They measured public facility evacuation times at train stations through their model, which produced predictions with a 15% mean accuracy. The system’s low accuracy level generates doubts about how credible it could be when used for critical emergency choices. Wang et al. [38] used random forest models together with Pathfinder simulation outputs to determine the evacuation times and exit selection choices of 4800 Xidi ancient architectural complex inhabitants. Researchers discovered that personal attributes determined which exits people would choose substantially more than surrounding environmental influences. The study depends on simulated information while studying only one complex structure, which affects its potential transferability to various settings. Yang et al. [39] executed a classroom violent attack evacuation study by applying random forest modeling to reach an accuracy rate of 96.5% in their findings. The SHAP analysis determined that distance to the attacker and preparation time were essential elements that affected decisions. The model delivers valuable knowledge about safety approaches, yet its dependent environment deviates from unpredictable emergency scenarios that exist in real-world settings.

Through transfer learning, Rahman et al. [40] adapted the DGCN-LSTM model, which had been trained on non-evacuation traffic for predicting network-scale evacuation traffic. The model proves suitable for evacuation traffic management, yet its performance involves barriers stemming from dependencies between features and beginning conditions. Without transfer learning, the model experienced difficulties processing evacuation data, while its functioning depends on particular features and becomes unstable during abrupt behavioral and scenario changes. Under dynamic circumstances, the model showcased prediction challenges through its RMSE value of 399.69. Research findings prove that deep learning and simulations deliver effective capabilities for predicting evacuation behavior. Li et al. [36], together with Clever et al. [37], developed CNN models for evacuation prediction, while Wang et al. [38] and Yang et al. [39] implemented random forest models on simulated and experimental data. Every research investigation has added meaningful findings, although implementation limitations emerge mainly from precision aspects and immediate operational capability, as well as adaptation to different building structures and emergency scenarios.

### 2.3. Smart and IoT-Based Emergency Navigation Systems

This section focuses on AI technology together with IoT devices, federated learning systems and reinforcement learning algorithms, which enhance real-time emergency route guidance and disaster recognition and evacuation schemes.

In contrast to Tan et al. [22], who proposed an emergency navigation system based on 3D modeling, IoT technologies, and edge computing with federated learning—achieving 99% accuracy in floor localization and 98.7% effectiveness in evacuation routing—our study focuses on integrating predictive fire detection with dynamic evacuation planning. While Tan’s system emphasizes post-incident route optimization, our approach aims to anticipate fire events before ignition, enabling proactive evacuation strategies in high-density environments like stadiums. Moreover, by leveraging environmental sensor data and deep learning, our framework addresses the limitations of traditional navigation systems that depend heavily on pre-configured infrastructure and may struggle with scalability. Yoo et al. [41] developed an indoor AR evacuation system through deep neural networks for position detection and reinforcement learning with gradient-boosting disaster prediction capabilities. The system demonstrates excellent performance by estimating areas and disaster propagation while offering instant directions to safety. The implementation faces two main disadvantages because it needs reliable beacon infrastructure along with strong IoT capabilities while also detecting real-time disruptions.

Using Kerala, India, flood victim questionnaire data, Sreejith et al. [42] constructed a machine learning model for assessing household flood preparedness duration. The research determined essential population characteristics and conduct patterns that affected people when they left their residences. Their model provides emergency planning knowledge through data analysis, yet its data collection method, which is based on self-reporting combined with its focus on preparatory stages, reduces its practical usefulness.

Unlike previous studies focusing solely on fire detection or evacuation modeling, this study presents an integrated framework combining predictive fire alarm systems with intelligent crowd evacuation strategies. The proposed EvacuNet model significantly advances predictive accuracy and reliability, achieving near-perfect performance metrics. Furthermore, this study evaluates environmental sensor data across a broader range of ecological parameters, offering a comprehensive and practical solution for large-scale venues like stadiums.

## 3. Methodology

The research methodology follows multiple essential steps, which include data collection, preprocessing the data before proceeding to the analysis stage, and model training before conducting evaluations, as represented in Figure 1. IoT devices collect data by monitoring environmental conditions to securely record different fire situation scenarios along with indoor and outdoor areas and multiple fire types under high humidity conditions. A total of 60,000 measurements form the dataset, which is sampled at 1 Hz.

### 3.1. Dataset Overview

The Smoke Detection Dataset [25] was selected in this study due to its suitability for training predictive models in environments resembling stadium conditions. Smoke Detection Dataset serves as a vital tool in our research about Crowd Evacuation in Stadiums Using Fire Alarm Prediction because it helps develop AI-driven smoke detection systems and improved fire safety responses. The two types of smoke detectors include photoelectric detectors, which detect light scattering from particles, and ionization-based detectors, which detect electrical disruptions from smoke. A data collection process focused on various locations was used to create a dependable AI-based smoke detection system that examined normal indoor and outdoor conditions, wood and gas fires in firefighter training zones, and outdoor grill configuration. The dataset includes about 62,630 sensor measurements with 15 features, which are recorded at 1 Hz for each UTC time-stamped entry. The combination of different environmental characteristics creates a thorough training set that produces accurate fire alarm predictions. The Smoke Detection Dataset provided a wide-ranging environmental scope because it collected fire sensor data inside and outside. The dataset does not include stadium data. It demonstrates representative potential fire patterns for big public areas through its diverse environmental conditions, including different humidity levels, CO_2_ concentrations, and fire types. This feature allows the model to obtain capabilities for real-world stadium applications. This research implements machine learning on IoT-based smoke detection to increase the speed of fire detection and reduce response times while enhancing stadium evacuation protocols for safe events.

### 3.2. Sensor Setup and Testing Conditions

The dataset utilized IoT-based environmental sensors, including photoelectric detectors (detecting light scattering) and ionization-based detectors (measuring electrical conductivity changes due to smoke). These sensors were calibrated using standardized fire training procedures. The data collection involved diverse conditions, including simulated wood fires, gas fires, and non-fire scenarios under various humidity levels, reflecting real-life emergency variability. Measurements were sampled at 1 Hz, ensuring high temporal resolution.

### 3.3. Data Preprocessing

The basic preprocessing methods for dataset quality control included duplicate value handling alongside null value management and statistical data analysis. We executed multiple steps to check for data errors along with missing values before declaring all sensor readings appropriate for model training purposes. We studied the dataset’s statistical parts through descriptive analysis to detect abnormalities and patterns in sensor data. The preprocessing techniques optimized the dataset, making it prepared for learning an accurate fire alarm prediction model through proper organization and removal of dataset inconsistencies.

### 3.4. Data Analysis

The research on stadium crowd evacuation focused on temperature distribution patterns and fire alarm activation to develop enhanced safety protocols, which are presented in Figure 2 and Figure 3. The data presented in Figure 3 illustrates two patterns: fire alarms appear most frequently in the 20–40 °C temperature range (red bar for yes responses), and the black bar for no responses has lower frequencies at these temperatures but shows a fire risk threshold showed by orange KDE curve. Two subplots divide the data sets in Figure 2, which demonstrates “Yes” alarm triggers occurring within the temperature range of 10–30 °C while “No” non-triggers exist primarily between 40 and 50 °C as reinforced by KDE curves. The visual graphs confirm how temperature functions as a critical factor in alarm operation and explain evacuation planning as well as sensor development.

Further, this study analyzed the distribution of Total Volatile Organic Compounds (TVOC) in relation to fire alarm triggers to enhance safety protocols, as depicted in Figure 4 and Figure 5. Figure 4 shows a stacked histogram of overall TVOC levels, indicating that lower TVOC values (0–2500) are strongly associated with fire alarms (“Yes” in red), while higher values (up to 60,000) are more common when no alarms are triggered (“No” in black). This suggests that TVOC thresholds are critical for fire detection. Figure 5 separates these distributions, with the “For Triggering Fire Alarm” subplot (red) peaking at low TVOC (0–1750) and the “For Not Triggering Fire Alarm” subplot (black) showing a broader distribution peaking at higher TVOC (10,000–60,000). These patterns, illustrated in Figure 1 and Figure 2, highlight the TVOC’s role in triggering alarms, informing evacuation strategies and sensor optimization in stadiums.

Again, this study analyzed CO_2_ equivalent concentrations during fire alarms in stadiums to develop improved safety standards, as presented in Figure 6 and Figure 7. The chart of accumulated CO_2_ readings displays two distinct patterns in Figure 6 that indicate the relationship between CO_2_ counts and fire alarm activation (Yes—red column) relative to detector silence (No—black column). Figure 7 divides the data into two groups, where the “For Triggering Fire Alarm” (red) distribution reaches its highest point at low CO_2_ amounts in the 0–5000 range, yet the “For Not Triggering Fire Alarm” (black) distribution achieves its maximum at higher CO_2_ values between 5000 and 40,000. Figure 6 and Figure 7 in the images demonstrate how CO_2_ plays a critical part in activating fire alarms in addition to leading stadium evacuation procedures and optimizing sensor detection capabilities.

The research on stadium crowd evacuation included a figure illustrating correlation findings about environmental factors and alarm triggers using Figure 8. A Pearson correlation heatmap shows the magnitude of variable relationships between Temperature, Humidity, TVOC, eCO_2_, Raw H2, Raw Ethanol, Pressure, PM1.0, PM2.5, NC0.5, NC1.0, NC2.5 and Fire Alarm. A close positive relationship exists between PM1.0 and PM2.5 and between their corresponding nanoparticle count versions NC0.5, NC1.0 and NC2.5 because each pair exhibits correlation coefficients approaching 1.0. A negative correlation viewed between Temperature and Humidity (−0.24) demonstrates opposing movements between the two variables, yet Fire Alarm shows low correlations with multiple variables, including weak associations between Temperature (−0.16) and Humidity (0.4). The presented analysis in Figure 1 uses data to reveal the main warning factors that contribute to fire risks, which helps develop stadium evacuation plans.

### 3.5. ML Models

In this paper, six different approaches of machine learning are used as tools for predicting fires and for evacuation planning. The existing models in these models are Logistic Regression, Gaussian Naïve Bayes, Bernoulli Naïve Bayes, Support Vector Machine (SVM) [6], Random Forest [7] and K-Nearest Neighbors (KNN). Each of these models was chosen as it can process various data patterns and may help to constitute meaningful fire detection and crowd evacuation strategies. The data were split 80:20 for training and testing purposes, while stratified sampling preserved class equilibrium. EvacuNet contained three dropout layers with L1 regularization to address overfitting issues and an early stopping mechanism that relied on validation metrics. The generalization: The matching alignment between training accuracy and validation accuracy curves and training loss curves and validation loss curves validates the generalization effectiveness of the model. This comparative analysis was conducted in order to identify the most effective machine-learning approach to improve fire safety and evacuation outcomes.

### 3.6. Proposed Method

The developed neural network model, EvacuNet, performs binary fire alarm classification on IoT sensor data through a deep learning structure featuring dense layers along with activation functions, regularization, and dropout strategies that enhance both learning and generalization capabilities. Each layer within the network operates under defined mathematical equations.

**First Dense Layer (Input to Hidden Layer 1):** The network starts at an input layer because the input features X of dimension d (i.e., the number of sensor readings) are transmitted through a Dense layer with 32 neurons. Each neuron applies a weighted additive operation before activating through the ReLU function.(1)$$Z^{\left(1\right)}=W^{\left(1\right)}X+b^{\left(1\right)}$$(2)$$A^{\left(1\right)}=ReLU\left(Z^{l}\right)$$
where $W^{1}$ represents the weight matrix, $b^{1}$ is the bias vector, and the ReLU activation is defined as:(3)$$ReLU\left(z\right)=\max\left(0,z\right)$$

**Regularization Term:** L1 regularization is applied to encourage sparsity, modifying the loss function by adding a regularization term:(4)$$L_{reg}=\lambda \sum \left|W\right|$$
where λ=0.003 controls the regularization strength.

**Second Dense Layer (Hidden Layer 1 to Hidden Layer 2):** The first hidden layer consists of 64 neurons, applying another transformation:(5)$$Z^{\left(2\right)}=W^{\left(2\right)}A^{\left(1\right)}+b^{\left(2\right)}$$(6)$$A^{\left(2\right)}=ReLU\left(Z^{\left(2\right)}\right)$$

**Third Dense Layer (Hidden Layer 2 to Hidden Layer 3):** By enlarging feature dimensions, the model can detect sophisticated data patterns. The second hidden layer implements 128 neurons, which enhances the feature extraction process.(7)$$Z^{\left(3\right)}=W^{\left(3\right)}A^{\left(2\right)}+b^{\left(3\right)}$$(8)$$A^{\left(3\right)}=ReLU\left(Z^{\left(3\right)}\right)$$

**Dropout Layer (Regularization):** To mitigate overfitting, a dropout layer is applied, randomly deactivating 30% of the neurons:(9)$$A_{dropout}^{\left(3\right)}=Dropout\left(A^{\left(3\right)},\;p=0.3\right)$$
where p is the dropout probability.

**Fourth Dense Layer (Hidden Layer 3 to Hidden Layer 4):** Following this, a third hidden layer with 16 neurons refines the feature space:(10)$$Z^{\left(4\right)}=W^{\left(4\right)}A^{\left(3\right)}+b^{\left(4\right)}$$(11)$$A^{\left(4\right)}=ReLU\left(Z^{\left(4\right)}\right)$$

**Output Layer (Binary Classification):** The final layer contains one neuron with a sigmoid activation function that provides probabilities between 0 and 1 at the output:(12)$$Z^{\left(5\right)}=W^{\left(5\right)}A^{\left(4\right)}+b^{\left(5\right)}$$(13)$$\hat{y}=\sigma \left(Z^{\left(5\right)}\right)=\frac{1}{1+e^{-Z^{\left(5\right)}}}$$
where σ(z) is the sigmoid function, ensuring that the output represents a valid probability score.

**Loss Function (Binary Cross-Entropy):** The model is trained using binary cross-entropy loss, defined as:(14)$$L=-\frac{1}{N}\sum_{i=1}^{N}\left[y_{i}log{\hat{y}}_{i}+\left(1-y_{i}\right)\log\left(1-{\hat{y}}_{i}\right)\right]$$
where $y_{i}$ is the true label and ${\hat{y}}_{i}$ is the predicted probability.

**Optimization Algorithm:** Optimization is performed using the Adam optimizer with a learning rate of 0.001, updating weights as:(15)$$W^{\left(t+1\right)}=W^{\left(t\right)}-\eta \cdot \frac{{\hat{m}}_{t}}{\sqrt{{\hat{v}}_{t}}+\in }$$
where: η = 0.001 is the learning rate, ${\hat{m}}_{t}$ and ${\hat{v}}_{t}$ are the bias-corrected first and second-moment estimates of the gradients, ∈ is a small constant to prevent division by zero.

This optimizer minimizes the binary cross-entropy loss efficiently while achieving overfitting reduction through dropout and regularization operations.

The Architectural view of the proposed model is shown in Figure 9.

## 4. Result and Discussion

This section presents the findings of the fire alarm prediction model and its impact on stadium evacuation strategies. The results demonstrate the effectiveness of machine learning in predicting fire hazards and optimizing emergency responses.

### 4.1. Fire Alarm Prediction Model Performance

A fire alarm prediction system evaluation used multiple machine learning models, which produced different performance results. The model based on Logistic Regression reached 89.54% accuracy, 0.89 precision, 0.90 recall, and an F1-score level of 0.89. Although the model showed successful results, it failed to reach maximum potential because other advanced models demonstrated superior performance (Table 1).

The Gaussian Naïve Bayes model showed the worst performance among all models, achieving an accuracy level of 76.37%, together with a precision of 0.76, recall of 0.77, and an F1-score of 0.72. The poor classification results indicate that features in this dataset do not exist independently and thus violate the underlying model assumption. Bernoulli Naïve Bayes offered reasonable results despite achieving an accuracy of 88.18% and precision, recall, and F1-score, all equaling 0.88, yet failed to become the optimal solution for fire alarm prediction.

Support Vector Machine delivered outstanding results compared to Naïve Bayes models by reaching an accuracy of 94.44% combined with a precision of 0.95 and recall of 0.94, and F1-score 0.94. High accuracy and precision outcomes from SVM resulted from successfully separating complex data points. The Random Forest technique delivered outstanding outcomes, resulting in a 97.01% accuracy rate as well as 0.96 precision and recall points and 0.98 F1-score points. The generalization abilities of Random Forest models indicate why this method proves to be an outstanding choice for prediction tasks regarding fire alarm systems.

K-Nearest Neighbors exhibited nearly flawless classification, largely due to its 99.93% accuracy rate, corresponding 0.9981 precision and complete recall, and 1.00 F1-score. The model demonstrates a highly effective capability to recognize fire alarms apart from non-fire alarms. KNN implements a complex calculation that requires significant computational resources, thus reducing its practical value in large-scale operational systems.

Among all the models tested, EvacuNet outperformed the rest, achieving an accuracy of 99.99% with perfect precision, recall, and F1-score values of 1.00. This result confirms that EvacuNet is the most effective model for fire alarm prediction, as it can learn complex patterns in the dataset with exceptional reliability.

The analysis of the results suggests that EvacuNet is the best model for predicting fire alarms due to its high accuracy and robustness. K-Nearest Neighbors also performed exceptionally well, but its computational cost may limit its practicality in real-time applications. Random Forest and SVM delivered strong results, making them viable alternatives for scenarios that balance accuracy and efficiency. In contrast, Naïve Bayes models (Gaussian and Bernoulli) had the weakest performance, indicating that simple probabilistic approaches are insufficient for this problem.

Compared to Lu et al. [10], whose best-performing model (AdaBoost) achieved an AuPRC of 0.78, our proposed EvacuNet reached an accuracy of 99.99% and an AUC near 1.0. Similarly, while Balboa et al. [13] reported an AUC of 0.831 using logistic regression for evacuation prediction, our model significantly outperformed this with perfect precision, recall, and F1-scores. These comparisons reinforce EvacuNet’s superior capacity in both predictive detection and evacuation optimization.

### 4.2. Analysis of the Confusion Matrix for EvacuNet Model

Figure 10 shows a graph showing the confusion matrix of the pinnacle model. The matrix displays precise analysis regarding how well the model predicts stadium evacuation fire alarms. The model achieved 3457 accurate predictions for cases with no fire alarm activity and 8931 hits for instances with fire alarm triggers. The model generated 137 incorrect positive predictions that showed fire alarm activation when no such event occurred but correctly identified only 8931 actual fire alarms out of the examined cases.

The modeled system exhibits substantial accuracy and reliability standards for detecting stadium fire alarms at an early stage. The model demonstrates superior identification capability of active fire threats primarily because of its low false negative output rate, which supports quick emergency response efforts and successful crowd evacuation. The model demonstrates excellent robustness because it accurately differentiates between alarm and non-alarm instances.

The occurrence of false positives implies that the model tends to mistake certain non-emergency conditions for actual fire alarms, thereby causing extra evacuations of people from the building. Such incidents that protect safety can trigger panic reactions and operational interruptions affecting resource distribution. Further model improvements through threshold optimization, feature selection, and boundary adjustment will help overcome the identified limitation. By optimizing hyperparameters and applying advanced ensemble methods, the model’s predictive capabilities would improve while maintaining its ability to detect actual threats. Implementing real-time sensor data collection alongside contextual information into predictive frameworks would create better operational decisions by maintaining safety levels in large-scale crowd environments. The outstanding performance of EvacuNet revealed the necessity to improve its operational accuracy due to 137 false alarm detections. These false alarm events could cause real stadiums to experience unnecessary panic situations and possibly require unnecessary stadium evacuations. Future work will enhance model precision by adding context-based filtering with a dynamic threshold system that maintains sensitivity. Implementing human-operated verification systems within the system can help reduce operational interruptions.

The EvacuNet model delivers outstanding results for fire alarm prediction, as indicated by its ROC curve performance shown in Figure 11. The curve shows a close-to-perfect threshold between positive and negative examples because the model obtained an AUC value approaching complete separation. The model demonstrates a high capability to recognize fire and non-fire events, so it produces few mistakes during classification. When the curve rises steeply from the beginning, it shows both a very low false positive rate (FPR) and a high actual positive rate (TPR). The model demonstrates high reliability during emergencies because it accurately detects fire incidents while avoiding unnecessary false alarms. EvacuNet performs better than random chance because its curve surpasses the diagonal gray line for both the train and test sets on both classes.

The high AUC value empowers the model for use in high-occupancy stadiums because it delivers dependable and swift fire detection, which is essential for maintaining public safety. The model effectively minimizes accidental alarms, and its high detection rate ensures proper emergency response initiation only when necessary.

EvacuNet is the superior model for fire alarm prediction based on ROC analysis results since it produces precise decisions with great confidence levels during real-time applications. Developers should focus on implementing this system within IoT-based fire detection systems to enhance stadium evacuation procedures.

### 4.3. ROC Curve Analysis of EvacuNet for Fire Alarm Prediction in Stadium Evacuations

The accuracy graph demonstrates that initial training values and validation accuracy remain minimal until they escalate dramatically during the first epochs. The model demonstrates rapid mastery of meaningful features from the dataset because of its accuracy. At this stage, the model changes its inner parameters to reduce prediction mistakes, thus leading to increased accuracy levels. The accuracy increases gradually throughout training because the model learns to comprehend the dataset more precisely. The accuracy reaches a point where it stays close to 100%, which signifies that the model acquired most of the essential patterns required for accurate predictions. The training and validation accuracy curves closely follow each other in this depiction (Figure 12).

Model generalization success for new data becomes evident through this observation. A significant difference between validation and training accuracy would indicate that the model memorized the training data instead of learning general patterns, representing classical overfitting. The model demonstrates equally high-performance levels across training and validation data according to the nearly matching accuracy curves. Model performance indicates no underfitting or overfitting since it maintains the best balance to achieve good results on previously seen data alongside new examples. The accuracy shows consistent gradual improvement after the fast learning period ends. The model demonstrates advanced learning capabilities through the ability to improve its interpretation of refined details within the provided information. The model has mastered all dataset patterns because accuracy reaches nearly 100% stability, indicating it does not benefit substantially from more training.

The loss graph delivers additional perspectives on how a model learns during training. A model’s performance quality improves as loss values decrease because it shows how accurately the predictions match the actual labels. During the early stages of training, both the training loss and validation loss demonstrate high initial levels since the model begins with random weights and has not encountered significant data patterns. The model enhances its predictive accuracy while training and validation loss values decrease rapidly during the learning process. A downward shift at such a steep rate shows that the optimization process operates well since parameter adjustments lead to reduced errors. Loss values reach stability after 40 training phases because the model has almost achieved its best solution; thus, additional training produces minimal improvements.

In this graph examination, the training loss and validation loss maintain a tight connection throughout the entire training process. The model demonstrates practical prediction accuracy improvements that carry over from training data into previously unseen information. An excessive focus on training data specializations, or overfitting, is indicated when the validation loss increases, yet the training loss keeps decreasing. Testing and training loss curves display almost identical patterns, suggesting that the model applies knowledge beyond training data memorization.

The validation loss exhibits minor fluctuations, yet these changes remain of limited magnitude without causing substantial instability. Deep learning models usually present such variations because their datasets feature complex examples that lead to unstable classification. The variations in validation loss stem from both the random elements of training and variations in sample batches, together with inherent noise in the data. The minor differences in validation loss do not affect the interpretation that the model is learning effectively without showing signs of overfitting.

The model displays excellent generalization properties through its accuracy and loss patterns. It shows proper learning rate adjustment due to an initial rapid learning period that transitioned into stable performance. The model maintains equivalent performance between its training and validation phases, indicating it will produce dependable outcomes for fresh data in practical settings. The model achieved nearly perfect final accuracy by successfully extracting the essential features from the well-structured dataset. Reaching this level of accuracy raises doubts about whether the dataset’s simplicity allows easy classifications. Real-life conditions make it challenging to get perfect accuracy because unseen data generally shows increased variability. Testing the model with new independent data would be beneficial to prove its stability range. The loss and accuracy curves indicate proper training execution and suitable model architecture selection, making it possible to assess whether the model falls between correct and incorrect fit areas. Implementing data augmentation and ensembling multiple models alongside hyperparameter optimization can enhance the application’s performance.

## 5. Conclusions and Future Directions

Stadiums with high population density need optimized fire detection systems and evacuation plans for public safety measures. Traditional fire alarm systems need to detect issues after they occur, before possible emergency responses, leading to longer evacuation times because they detect late. The research aims to solve current problems through predictive fire alarm technology, incorporating optimized evacuation solutions to improve emergency response effectiveness. The research reveals that monitoring environmental sensors in real-time demonstrates how predictive AI models create rapid-fire alerts, leading to better evacuation procedures to cut risks in fire situations. The main conclusion of this investigation reveals that EvacuNet operates as the most effective method for forecasting fire hazards. EvacuNet displayed superior performance above Logistic Regression and Support Vector Machines (SVM), Random Forest, and K-Nearest Neighbors since it captured 99.99% accuracy with F1-score, precision, and recall at 1.00. According to research findings, EvacuNet demonstrates an outstanding capability to identify genuine fire risks from false alarms, thus surpassing standard fire detection systems. False alarms must be reduced in high-occupancy premises because unplanned evacuations result in financial damage, panic, and operational disruptions. The system lowers false alarm occurrences, guiding emergency crews toward relevant security threats.

The research incorporates predictive fire detection methods alongside real-time evacuation planning approaches as its primary academic finding. Through the application of machine learning for crowd movement optimization, the overall evacuation time decreased, and this reduction eliminated congestion that could have led to stampedes. According to this research, real-time fire predictions and adaptive evacuation modeling systems create a congestion prevention system that leads to improved structured and efficient evacuation procedures. The identified findings serve as essential resources for disaster managers, stadium operators, and emergency response teams who depend on data-centric decision systems to enhance public safety operations. According to these findings, predictive fire alarm systems become more effective by incorporating environmental sensor data. The examination included 15 essential ecological measurements from 62,630 sensors, which monitored temperature and humidity and tracked TVOC levels, CO_2_ emissions, and PM amounts. Such relationships establish the requirement for IoT-based fire detection systems using real-time monitoring to detect developing fire hazards. The deployment of AI-enhanced IoT sensors throughout stadiums and vast public buildings will enhance fire safety, shorten emergency response durations, and boost emergency preparedness. The study reveals extensive consequences of machine learning use in public safety domains. Research has created fundamental principles for upcoming disaster response automation by combining AI fire prediction technology with inventive evacuation systems. The adoption of this solution faces particular obstacles because of its computational needs, difficulty in understanding processor frameworks, and public safety restrictions. Additional research needs to direct efforts toward four critical areas: improving real-time adaptability through reinforcement learning algorithms and model expansion to support different emergency types, including earthquakes and chemical spills.

The research shows that fire alarm prediction and intelligent evacuation systems using AI technology create superior public security measures in high-volume facilities such as stadiums. Real-time sensor monitoring combined with machine learning models and predictive analysis tools allows this method to slash false alarm occurrences and enhance evacuation performance while permitting proactive emergency action. Predictive fire detection systems combined with adaptive crowd management methods create organized evacuation procedures that prevent overcrowding while lowering the number of injuries during emergency responses. Future investigations in emergency response technology must develop IoT systems to partner with adaptive evacuation models trained through reinforcement learning, which alters their response based on dynamic environmental shifts. Research should develop multi-agent simulation frameworks for optimizing large-scale evacuation processes through earthquakes, chemical hazards, and terrorist incidents. Increased deployment of 5G infrastructure alongside edge computing capabilities and automated AI-controlled emergency systems will optimize situational awareness, thus leading to prompt and more precise decision-making. The implementation of AI-enhanced predictive fire detection in combination with smart evacuation frameworks is an essential requirement for reducing time in emergencies, casualty prevention, and enhanced urban sustainability. Modern emergency management systems enhanced by sensor networks, smart navigation systems, and crowd analytics tools will better protect lives in dense urban spaces.

## Figures and Tables

**Figure 1 sensors-25-02810-f001:**
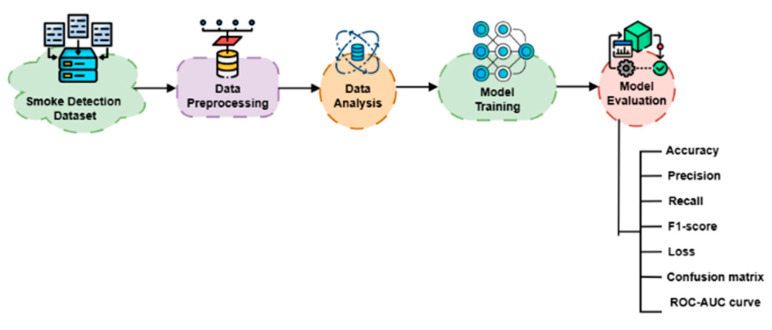
The overall workflow of this study.

**Figure 2 sensors-25-02810-f002:**
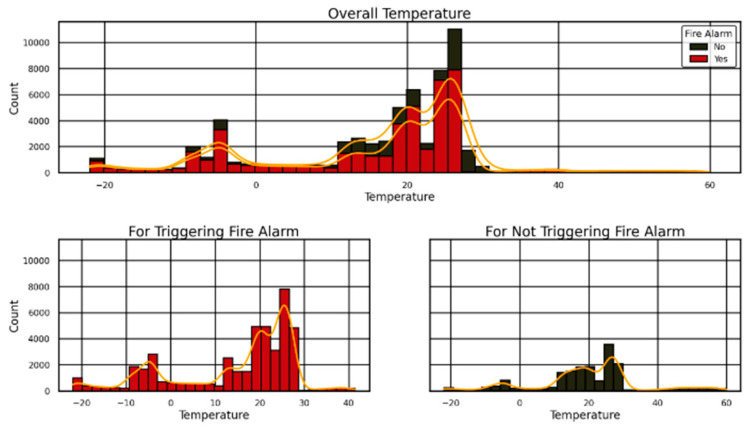
Temperature histograms for stadium fire alarms (“Yes” in red) and non-triggers (“No” in black), with orange KDE for evacuation insights.

**Figure 3 sensors-25-02810-f003:**
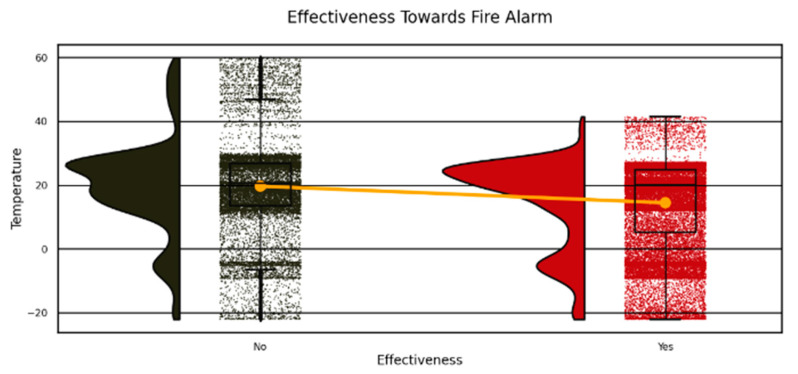
Stadium temperature histogram: fire alarms (“Yes” in red) vs. non-triggers (“No” in black), with orange KDE for evacuation risk.

**Figure 4 sensors-25-02810-f004:**
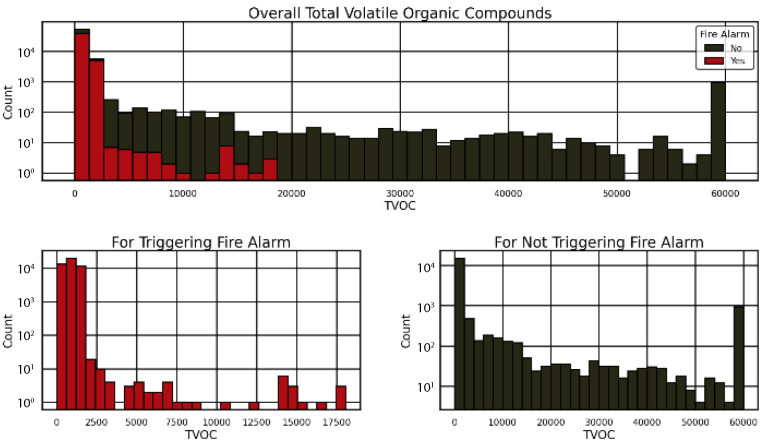
Histograms of TVOC for stadium fire alarm triggers (“Yes” in red) and non-triggers (“No” in black), revealing patterns for evacuation decision-making.

**Figure 5 sensors-25-02810-f005:**
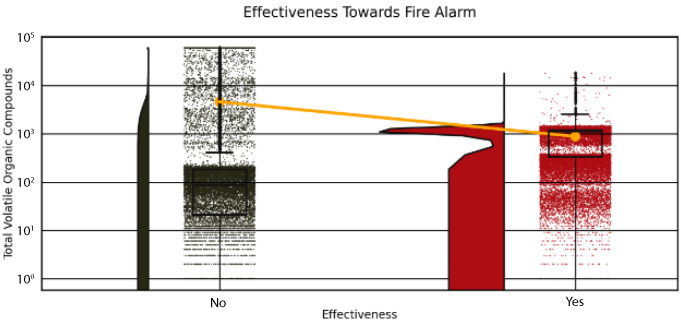
A stacked histogram of stadium TVOC distribution shows fire alarm triggers (“Yes” in red) and non-triggers (“No” in black), indicating TVOC’s role in fire detection for evacuation.

**Figure 6 sensors-25-02810-f006:**
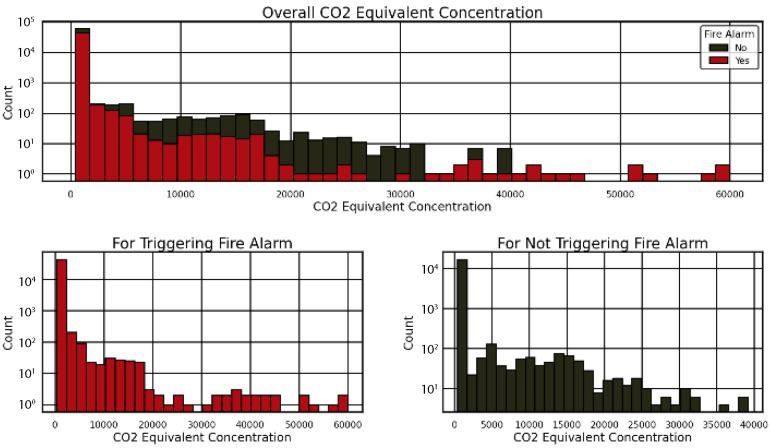
Histograms of CO_2_ equivalent for stadium fire alarm triggers (“Yes” in red) and non-triggers (“No” in black), revealing patterns for evacuation decision-making.

**Figure 7 sensors-25-02810-f007:**
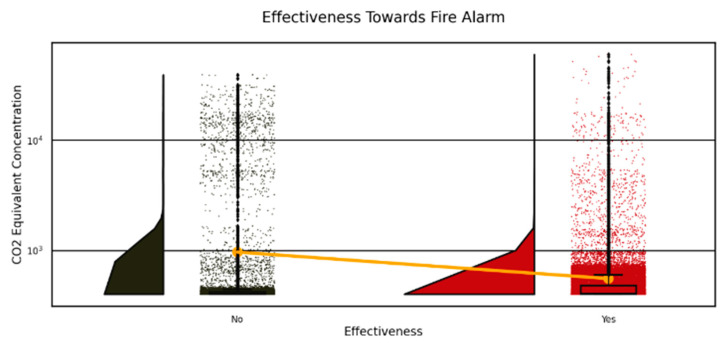
A stacked histogram of the stadium’s CO_2_ equivalent concentration shows fire alarm triggers (“Yes” in red) and non-triggers (“No” in black), highlighting CO_2_’s role in fire detection for evacuation.

**Figure 8 sensors-25-02810-f008:**
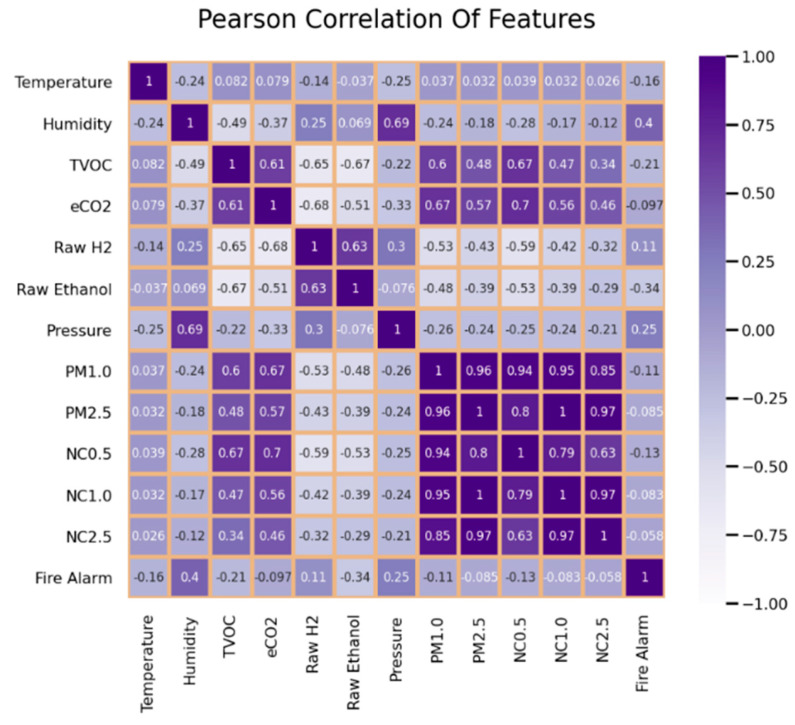
Pearson correlation heatmap of environmental features.

**Figure 9 sensors-25-02810-f009:**
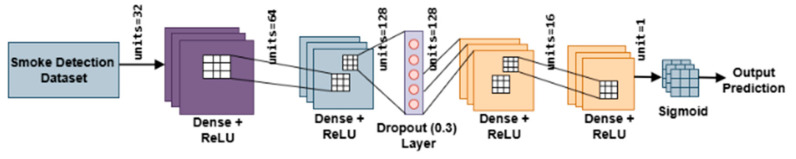
Architecture of the proposed model.

**Figure 10 sensors-25-02810-f010:**
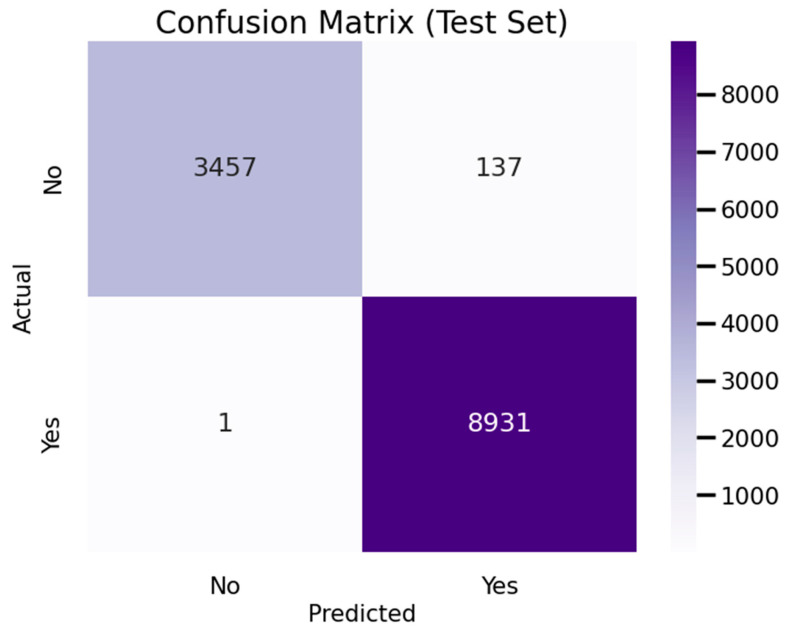
Confusion Matrix for EvacuNet Model.

**Figure 11 sensors-25-02810-f011:**
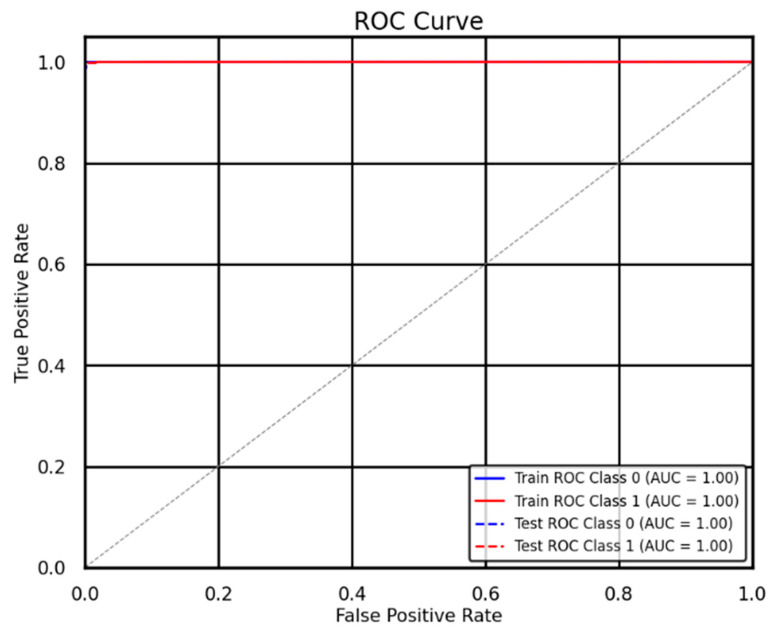
ROC Curve for EvacuNet Model.

**Figure 12 sensors-25-02810-f012:**
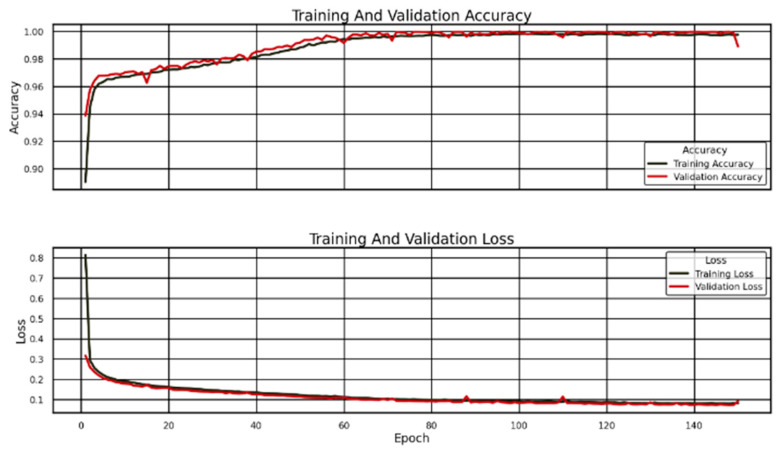
Model Performance: Accuracy and Loss Over Training Epochs.

**Table 1 sensors-25-02810-t001:** Model Performance Comparison.

Model	Accuracy (%)	Precision	Recall	F1-Score
Logistic Regression	89.54%	0.89	0.90	0.89
Gaussian Naïve Bayes	76.37%	0.76	0.77	0.72
Bernoulli Naïve Bayes	88.18%	0.88	0.88	0.88
Support Vector Machine	94.44%	0.95	0.94	0.94
Random Forest	97.01%	0.96	0.96	0.98
K-Nearest Neighbors	99.93%	0.9981	1.00	1.00
EvacuNet (Best Model)	99.99%	1.00	1.00	1.00

## Data Availability

Data is contained within the article.

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
