# Peer review of "Crowd Evacuation in Stadiums Using Fire Alarm Prediction"

_sensors, 2025, doi:10.3390/s25092810_

Round 1
Reviewer 1 Report
Comments and Suggestions for Authors
This paper investigated an AI-driven predictive fire alarm and evacuation model that leverages machine learning algorithms and real-time environmental sensor data to anticipate fire hazards before ignition, improving emergency response efficiency. However, there are some issues that need to be addressed.
In the literature review section, it is necessary to explain the characteristics of this study.
The format of the references is not standardized, such as [22].
The reason that this study selected the goal of the smoke detection dataset [25].
Is the proposed method first proposed or has it been used by previous researchers?
Comments on the Quality of English LanguageThe English could be improved to more clearly express the research.
Reviewer 2 Report
Comments and Suggestions for Authors
The article presents a significant and compelling contribution to the field of public safety in high-density environments such as stadiums. The research addresses the critical problem of the limitations of traditional reactive fire alarm systems and proposes a proactive solution based on artificial intelligence.
Despite the contribution of the study, there are certain shortcomings in the text that would be appropriate to modify within the review process. I recommend conducting a revision in the area of methodology, whereby it is necessary to describe the applied approach and its verification in more detail. It would also be appropriate to supplement and update the professional references and expand the interpretation of the results, especially their comparison with existing solutions.
Finally, it is important to ensure the consistency and completeness of citations, while adhering to a unified citation style – in accordance with the guidelines of the relevant journal.
In the methodology section, it would be appropriate to include a detailed description of the devices used, testing conditions, sensor calibration, the number of measurement repetitions, and the method of result evaluation. The dataset contains 62,630 sensor measurements across 15 parameters, representing an extensive dataset. The article states that the data were collected under typical indoor and outdoor conditions; however, it is not explicitly mentioned whether this dataset sufficiently represents the diverse scenarios and types of fires that may occur in a real stadium environment. To enhance the robustness of the analysis, it would be beneficial to include information on the variability of conditions and their relevance to the simulated situations.
The confusion matrix for the EvacuNet model reports 137 false positive predictions, which can lead to unnecessary evacuations, panic reactions, and disruption of operations. The article acknowledges this issue, but for the practical implementation of the model in a real-world environment, it is crucial to minimize the occurrence of false alarms. Although the rate of false negative results appears to be low, emphasis should be placed on optimizing the detection algorithm in order to reduce the number of incorrect positive predictions and ensure higher system reliability.
In the area of professional references, the document cites relevant literature, but newer studies (e.g., from 2022-2024) that focus on modern prediction and simulation methods could be included. I recommend adding references to current studies and scientific papers and citing specific standards or technical specifications due to the lack of citations for some technical data and claims regarding sensor effectiveness.
Overall, I believe that this research is very promising. The development and successful validation of the EvacuNet model, along with a focus on integrating fire prediction with evacuation planning, represent significant progress in the use of AI for public safety in densely populated areas. The study provides compelling evidence of the potential for predictive fire signaling systems to surpass traditional reactive systems by increasing detection speed, reducing false alarms, and optimizing evacuation procedures. The identified future research directions suggest a clear path toward even more sophisticated and adaptive emergency response systems. Nevertheless, it would be beneficial to include more empirical data and conduct a more detailed analysis of the reliability of the models used.
Reviewer 3 Report
Comments and Suggestions for Authors
This study proposes an AI-driven fire prediction and evacuation optimization framework (EvacuNet) for crowded stadiums. comments are as follows:
1.Data splitting strategy is unspecified.
2.EvacuNet’s near-perfect accuracy (99.99%) raises concerns about overfitting or data leakage.
3.The literature review section suggests reorganizing to highlight the innovativeness of this research.
4.Test results can be compared with previous models proposed by other scholars.
Round 2
Reviewer 1 Report
Comments and Suggestions for Authors
The English is fine and does not require any improvement.
Reviewer 3 Report
Comments and Suggestions for Authors
I recommend accepting this paper.